# Model for Hydrogen Production Scheduling Optimisation

**Vitalijs Komasilovs** [1], **Aleksejs Zacepins** [1,*], **Armands Kviesis** [1] **and Vladislavs Bezrukovs** [2]

1　Institute of Computer Systems and Data Science, Faculty of Engineering and Information Technologies, Latvia University of Life Sciences and Technologies, Liela iela 2, LV-3001 Jelgava, Latvia; vitalijs.komasilovs@lbtu.lv (V.K.); armands.kviesis@lbtu.lv (A.K.)

2　Engineering Research Institute "Ventspils International Radio Astronomy Centre" (ERI VIRAC), Ventspils University of Applied Sciences (VUAS), Inzenieru Str. 101, LV-3601 Ventspils, Latvia; vladislavsb@venta.lv

*　Correspondence: aleksejs.zacepins@lbtu.lv

**Abstract:** This scientific article presents a developed model for optimising the scheduling of hydrogen production processes, addressing the growing demand for efficient and sustainable energy sources. The study focuses on the integration of advanced scheduling techniques to improve the overall performance of the hydrogen electrolyser. The proposed model leverages constraint programming and satisfiability (CP-SAT) techniques to systematically analyse complex production schedules, considering factors such as production unit capacities, resource availability and energy costs. By incorporating real-world constraints, such as fluctuating energy prices and the availability of renewable energy, the optimisation model aims to improve overall operational efficiency and reduce production costs. The CP-SAT was applied to achieve more efficient control of the electrolysis process. The optimisation of the scheduling task was set for a 24 h time period with time resolutions of 1 h and 15 min. The performance of the proposed CP-SAT model in this study was then compared with the Monte Carlo Tree Search (MCTS)-based model (developed in our previous work). The CP-SAT was proven to perform better but has several limitations. The model response to the input parameter change has been analysed.

**Keywords:** hydrogen; OR-Tools; cost optimisation; operation scheduling; electrolyser scheduling; constraint programming





## 1. Introduction

The relentless expansion of both the global population and economy, coupled with the transition from rural to urban lifestyles, has resulted in a substantial surge in the demand for energy [1]. Promoting the use of renewable energy (RE) sources brings important advantages [2], like the reduction in carbon dioxide ($CO_2$) emissions and is an important step towards energy sustainability. One of the scenarios for future energy sustainability is the production of "green" hydrogen ($H_2$). Hydrogen is an energy carrier, and its "cleanness" directly reflects the energy source and methods that were used in its production [3]. To classify hydrogen as "green" its production must not involve any (or minimal) greenhouse gas emissions. One way to achieve this is to solely base the production on RE sources [4].

The production of "green" hydrogen is based on water electrolysis where the water is split into two gases—$H_2$ and oxygen ($O_2$). One of the sustainable methods of water splitting is the use of a proton exchange membrane (PEM) water electrolyser, which is powered by RE sources [5]. To fully power a PEM electrolyser from renewable energy sources, however, introduces a lot of challenges related to the availability of RE affected by day–night cycles, seasonality and other factors (e.g., climate change). This directly impacts the operational stability of the electrolyser. Therefore, a combination of RE sources and electrical energy from the grid [6] can be a valid solution to power the electrolyser and avoid intermittency periods. However, in this case, the produced hydrogen may not be

considered "truly green" due to the fact that the source of the produced energy in the grid is unknown.

The combination of various energy sources requires a control strategy that manages the scheduling and state switching of the electrolyser. The scheduling problem in areas related to the industrial sector has been studied since the last century [7,8].

In industrial applications, scheduling problems that pose computational challenges are widespread [9]. Some examples of such problems are staff rostering [10], hoist scheduling [11], satellite scheduling [12], etc. Regarding hydrogen production, [13] proposed a control strategy for the sequential operation of multiple PEM electrolysers, depending on the availability of photovoltaic power.

Electrolyser optimisation can partially be considered as a single-machine scheduling problem. Given a single machine (e.g., device or shop) and the number of variable length jobs (e.g., tasks or operations) to be executed on the machine, the problem is to schedule the jobs in such a way that optimises specific objectives (e.g., minimises costs or maximises throughput). In computer science and operations research, such a problem is referred to as single-machine scheduling and is usually related to variable length jobs and optimisation of critical deadlines, while in the case of electrolyser operation, it is more an assignment problem where a "job" should be assigned for each time span, but with existing specific constraints on job sequences (thus not purely assignment).

Previously, we developed an MCTS (Monte Carlo Tree Search) model for hydrogen production optimisation [14], but it was found that the model is not able to find a truly optimal solution in very deep trees and thus it is necessary to search for other approaches for solving this problem. While MCTS has proven to be successful in various applications, it also comes with certain limitations and challenges, such as limited exploration in early stages and it may struggle to explore the search space effectively in the early stages of the search [15].

Another approach for the hydrogen production optimisation problem is to formulate it as a mathematical model and analyse it using general purpose solvers. Constraint programming is particularly well suited for this problem. Constraint solving plays a vital role in different kinds of problem analysis and optimisation tasks and has the ability to be more efficient compared to other approaches [16,17].

The efficiency of hydrogen electrolysers can be greatly improved by applying appropriate control strategies [18]. The implementation of such strategies is crucial because various parameters, like the frequency of electrolyser on/off cycles, influence the overall system performance [14].

The aim of this research was to develop a model to improve the control of $H_2$ electrolyser operation, considering the combination of various energy sources and varying electricity prices and to tackle the disadvantages of the previously developed MCTS model.

## 2. Materials and Methods

### 2.1. Optimisation Problem Setup

An electrolyser schedule optimisation problem can be formulated as a combination of optimisation problems: (a) single-machine scheduling problem (class of optimal job scheduling problems) and (b) assignment problem (class of combinatorial optimisation problem). The optimisation goal is to maximise revenue from $H_2$ and $O_2$ production subject to varying grid energy prices and availability of RE sources. In our particular case, various electrolyser states (offline, various levels of production, startup or shutdown sequences) with fixed duration and variable cost profiles are assigned to each timespan taking into account defined state transition constraints. An inherent trait of electrolysers is their process inertia, where the initiation and cessation of $H_2$ production do not occur immediately upon activating or deactivating the electrolyser. Electrolysers undergo distinct "StartUp" and "Shutdown" sequences, which may extend up to 1 h, as employed in our model. The duration of these sequences may differ among various electrolysers, influenced

by factors such as the necessary electrolyte temperature for optimal efficiency and the types of membranes employed.

The model incorporated specific state sequences along with their corresponding energy consumption profiles. The model also takes into account some fixed costs that represent the degradation of internal components and the associated maintenance.

Given directional graph of electrolyser states *S* and transitions *Tr* between them $G = \{S, Tr\}$, the optimisation goal is to maximise cumulative profit over time horizon *T* as follows (see Equation (1)):

$$\max\left\{\sum_{t\in T} F(t, s) : s \in S, s_t \in G^+(s_{t-1})\right\} \tag{1}$$

where $G^+$ denotes sub-graph of states accessible from the previous state.

We employed a simplified hydrogen production model described in detail in [14]. The model has three discrete production states, expressed as low, medium and high. We selected a small-scale hydrogen electrolyser from the market (e.g., https://pureenergycentre.com/hydrogen-products-pure-energy-centre/hydrogen-electrolyser/ (accessed on 8 February 2023)), to base our assumptions on its production rates and power, like the one with an hourly H2 production of 1 kg with maximum operating power of ~60 kW. In this study, the electrolyser model and objective function stay the same to have comparable results with the MCTS approach.

The profit for specific timestamp and electrolyser state is calculated as follows (see Equation (2)):

$$F(t,s) = Q_s^{H2} \times P^{H2} + Q_s^{O2} \times P^{O2} - \max\left(E_s - E_t^{PV}, 0\right) \times P_t^E - C_s \tag{2}$$

Time horizon for the optimisation problem is set to 24 h due to the peculiarities of the grid energy market. In particular, Nordpool power exchange platform offers a day-ahead market (https://www.nordpoolgroup.com/ (accessed on 12 December 2023)), thus the energy prices are fixed only for the next day. In this study, we use spot market prices for the 8th of February, 2023 for result comparison reasons. Also, there is potential for integrating models for predicting grid energy prices and/or RE sources for extended periods, but these models are not in the scope of this research. Energy prices and RE are input parameters for our model, thus fixed input values for a specific date can be substituted by outputs of other models and/or online services. The model prioritises the available RE and only covers the remaining energy demand from the grid.

Table 1 below summarises all the variables used in research and calculations:

**Table 1.** List of nomenclature.

| Symbol | Property | Units |
|:---:|:---:|:---:|
| $Q_s^{H2}$ | Quantity of H$_2$ produced in a specific state | kg |
| $Q_s^{O2}$ | Quantity of O$_2$ produced in a specific state | kg |
| $P^{H2}$ | Price of H$_2$ | EUR/kg |
| $P^{O2}$ | Price of O$_2$ | EUR/kg |
| $P_t^E$ | Price of electricity | EUR/kWh |
| $E_s$ | Energy consumed in a specific state | kWh |
| $E_t^{PV}$ | RE available in a specific time span | kWh |
| $C_s$ | Amortisation costs in a specific state | EUR |
| $F$ | Profit | EUR |

The following components that are used in the model are the same as described in our previous work [14]:

- An array of photovoltaic solar (PV) cells with a theoretical maximum capacity of 100 kW. This represents the source of RE with available excess power that can vary over the day. Other RE sources can be considered as well.
- Relative solar irradiation is simulated for daylight periods corresponding to the spring or fall equinoxes. Specific latitudes and the day of the year have further effects on solar declination angle and power output of PV cells.
- The power grid connection is considered a backup energy source. Its capacity is not limited, but it has fluctuating prices for each hour.
- The main income for the electrolysis process is modelled as sales of the output products ($H_2$ and $O_2$). While prices of products fluctuate depending on production methods, required post-processing and targeted applications, the model considers fixed prices of the products: $PH_2$ for EUR 5.00/kg and $PO_2$ for EUR 0.10/kg.

### 2.2. Constraint Satisfaction Problem Optimisation Using OR-Tools

The defined optimisation problem is well suited for constraint programming techniques and can be reduced to a constraint satisfaction problem. We used an OR-Tools software suite for analysing the problem, in particular CP-SAT solver [19].

The open-source software suite OR-Tools (developed by Google's Operations Research team) contains several solvers designed to solve optimisation problems of various kinds. One of the solvers is the mentioned CP-SAT, which employs satisfiability methods for solving constraint programming problems [20]. CP-SAT solver can be applied in many industries and for solving different scheduling problems. For instance, it was used for satellite scheduling problems [21], vehicle routing problems [22–24], shift scheduling in healthcare [25], hoist scheduling [26], recharging scheduling of electric buses [27] and many more.

In constraint programming, constraints are used to express relationships between variables in a problem. Constraints can be classified into two main types: hard constraints and soft constraints. Hard constraints are mandatory conditions, domain constraints and precedence constraints. Soft constraints are preference conditions, resource utilisation preferences, cost constraints and flexibility constraints.

The following constraints were implemented for the electrolyser scheduling model (see Table 2).

**Table 2.** Constraints used in the scheduling model.

| Constraint Formulation | Description |
|:---:|:---|
| $n(s_t) = 1, \forall t \in T$ | Exactly one state assigned to each timestamp. |
| $s_0 \in \{Off, StartUp\}$ | Initial state of the electrolyser is either *Off* or *StartUp* (similar to experiments with MCTS model). |
| $s_t \in \{s : s \in G^+(s_{t-1})\}, \forall t \in T$ | For all timespans, the state is selected from a subset of states, which are reachable from the previous state. |
| $max\{v \times F, \forall t \in T, \ \forall s \in S\}$ | The model variables are restricted to Boolean values indicating the state assignment to the timespan. The objective of the model is to maximise dot product between assignment variables and their corresponding revenue. |

The model and further experiments were implemented using Python 3.10 and corresponding interface for the OR-Tools library, specifically, we used *CpModel* and *CpSolver* from the *ortools.sat.python.cp_model* package.

The chosen parameters are not sourced from existing literature but instead are our assumptions by taking into account the selected electrolyser (mentioned previously in section "Optimization problem setup").

## 3. Results and Discussion

The following chapters describe experiments with the CP-SAT model and corresponding results: (a) the CP-SAT model was compared with the MCTS model on a similar optimisation problem and (b) the CP-SAT model response on various input parameters was analysed. All optimisations were performed for a 24 h time horizon with two time (15 min and 1 h) resolution settings. All figures were produced by the same Python script used to define and run the developed model.

### 3.1. CP-SAT and MCTS Model Comparison

Similar to the MCTS model, the CP-SAT model was optimised for a 24 h schedule with 1 h resolution and starting time $t_0$ at midnight with the electrolyser being switched off (thus the first valid states are either continued *Off* state or the beginning of *StartUp* sequence). Figure 1 shows the comparison between a manually created "full power" schedule, the results from a previously developed [14] MCTS algorithm and the best solution found by the CP-SAT model. The model outputs are as follows: total energy consumption $E$, hydrogen produced $Q_{H2}$, oxygen produced $Q_{O2}$ and profit $F$.

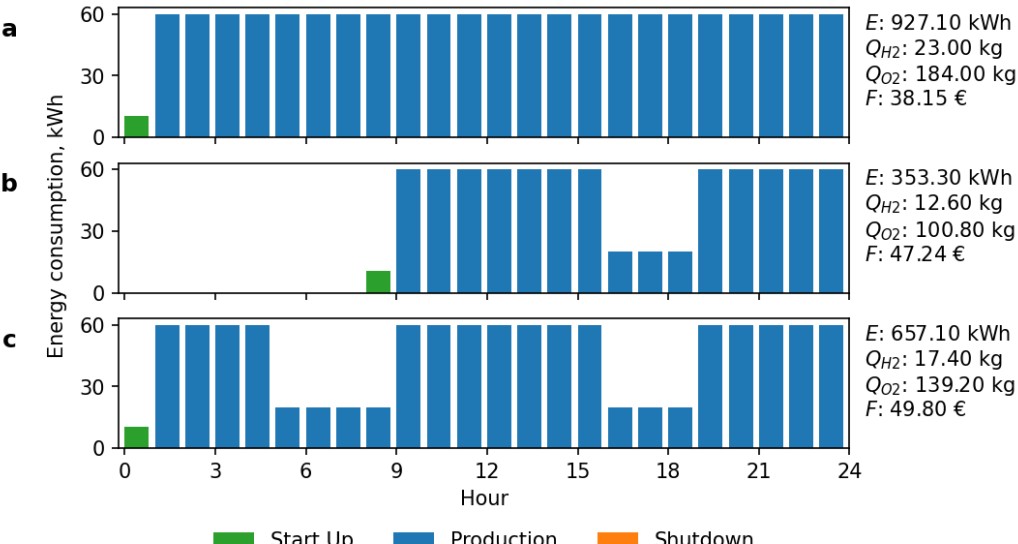

**Figure 1.** Electrolyser schedules compared (1 h resolution): (**a**) "full power" production, (**b**) the best schedule by MCTS and (**c**) the best schedule by CP-SAT model.

Despite almost doubling energy consumption, the CP-SAT model found slightly better cumulative profit for a given energy price profile. The results were cross-validated between MCTS and CP-SAT models to exclude bugs and discrepancies in profit calculations. Both models yielded exactly the same numbers on the same schedules.

A similar comparison was performed for electrolyser schedules with 15 min resolution. Figure 2a–c show the comparison between a manually created "full power" schedule, the results from a previously developed MCTS algorithm and the best solution found by the CP-SAT model. Again, despite increased energy consumption, the CP-SAT results have a slight edge over the MCTS in terms of overall profit.

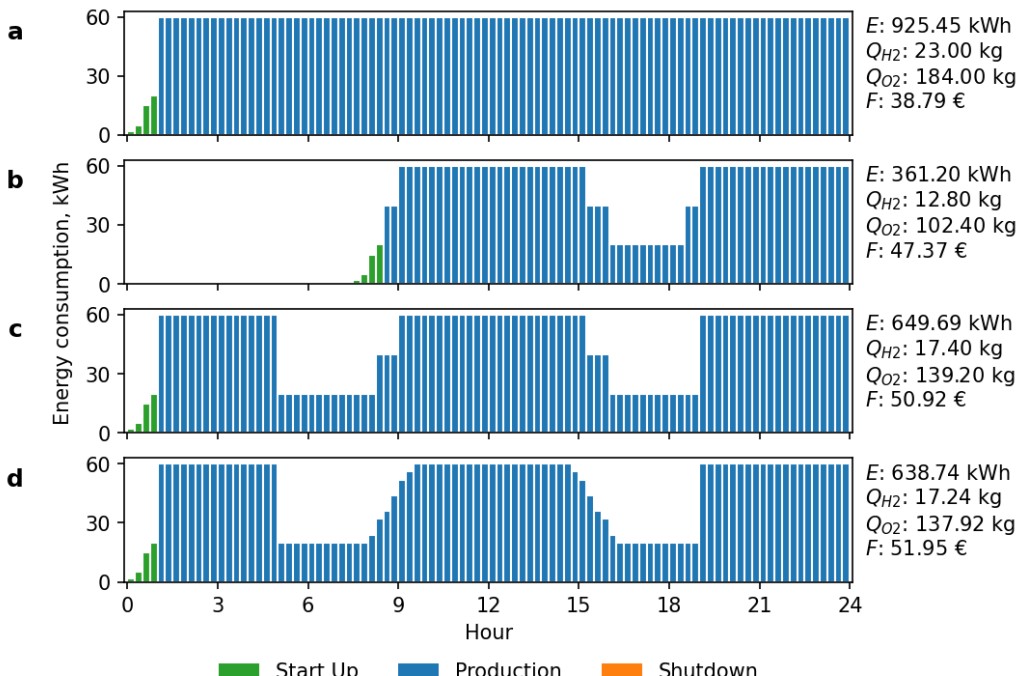

**Figure 2.** Electrolyser schedules compared (15 min resolution): (**a**) "full power" production, (**b**) the best schedule by MCTS, (**c**) the best schedule by CP-SAT model (fixed production levels) and (**d**) the best schedule by CP-SAT model (variable production levels).

The schedules produced by the CP-SAT model have a seemingly symmetrical energy consumption profile around midday. However, it is not affected by the model, but rather by available RE sources and grid electricity prices. At midday, the majority of energy demand is covered by solar panels, while during the night hours, lower grid electricity prices promote hydrogen production. The MCTS model has a non-symmetrical schedule due to its inability to find a global optimal solution. Only a very specific schedule starting at 00:00 yields profit, thus this solution space branch was rejected by MCTS in favour of solution candidates starting at 07:00.

In addition to discrete low, mid and high electrolyser production rates corresponding to 20, 40 and 60 kWh of power, we extended the CP-SAT model with variable production rates keeping the same lower and higher energy consumption bounds (see Figure 2d). In this case, the schedule had a better fit for the profile of available RE sources (solar power) and even better overall profit.

The results demonstrate that the CP-SAT solver overcomes the MCTS algorithm and finds a better schedule of the electrolyser in both cases (1 h and 15 min). In 1 h time resolution, the results are better, e.g., a profit increase of 5%, but in 15 min time frames, ~7.5%. If the model operates with variable production levels, then the result is improved by 2% more.

Also, the results correspond to the general electrolyser utilisation approach used by many companies, which implies non-stop operation of the device. In practice, this is often defined by the price of the electrolyser and expectations for the return on investments.

However, the model adopts production rates to the available energy and price profiles, thus opening the potential for higher profit.

### 3.2. Input Parameter Analysis for the CP-SAT Model

The electrolyser operational cost evaluation model relies on several external input parameters, such as grid energy prices, the availability of RE and maintenance costs. We performed a number of experiments with the CP-SAT model in order to reveal the response dynamics on changes in the input parameters and compared them to the baseline model described in the previous chapter.

The major position of the electrolyser cost profile is grid energy price and lowering it will increase the profit from hydrogen production. However, the electrolyser operator has no influence on grid energy prices, thus we consider it as an external force and unavoidable spending.

Another significant position in the cost/profit profile of the electrolyser is amortisation costs. These costs express the need to eventually replace electrolyser parts (mainly membrane) and are formulated in the form of fixed costs for the on–off cycle. Electrolyser components are susceptible to degradation mostly during the startup and shutdown phases. Due to relatively high amortisation costs it is preferable to keep the electrolyser in the operational state (referred to as "always on" schedule), thus lowering component wear. Figure 3 shows the effect of decreased amortisation costs (effectively, the cheaper electrolyser parts) on the schedule and overall profit: lowering costs of the on–off cycle by 50% and 75% results in higher profit by 3% and 10%, respectively.

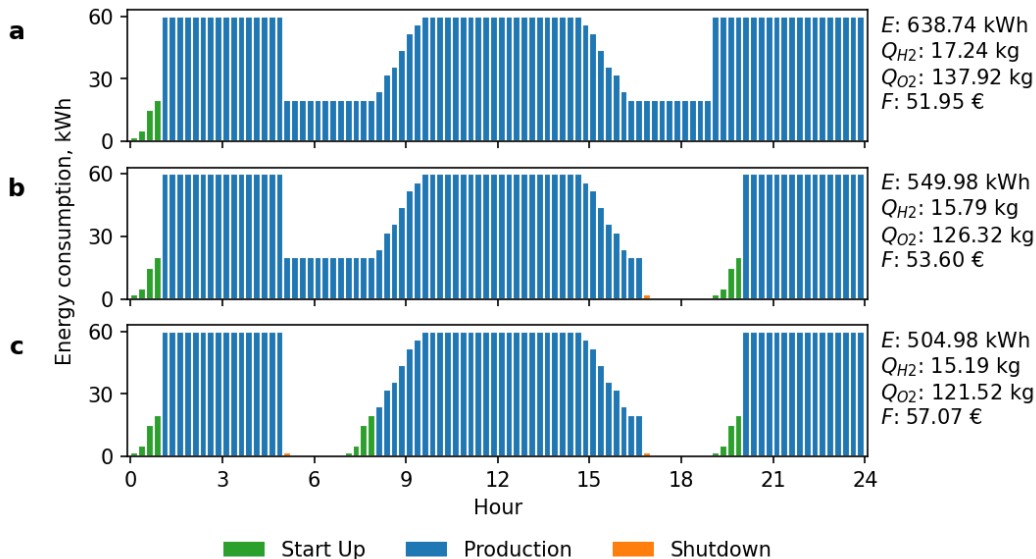

**Figure 3.** Electrolyser schedules depending on various amortisation costs per on–off cycle: (**a**) EUR 6.00 (baseline), (**b**) EUR 3.00 and (**c**) EUR 1.50.

This clearly demonstrates that the amortisation costs are the factor forcing an "always on" electrolyser schedule, while cheaper and/or less susceptible to wear technology would open possibilities for more sophisticated responses to such external factors, such as weather conditions and RE sources.

Figure 4 shows how the amount of available RE sources affects the electrolyser schedule. We evaluated scenarios with decreased peak power of the solar panels down to 60%, 30% and 10% from the baseline, which in practice can be interpreted as lower capacity or damaged solar panels, overcast days or the winter season. The corresponding profit is reduced down to 74%, 30% and 22% from the baseline.

At some point, electrolyser operation becomes unprofitable without RE sources. The last scenario (Figure 4d) demonstrates that a low level of solar power leads to idle schedules during the day, while the model still finds the opportunity for slight profit during the night hours when the grid electricity prices are the lowest. The negative impact of this key factor on the production process can be reduced via advanced forecasting techniques and/or additional sources of RE.

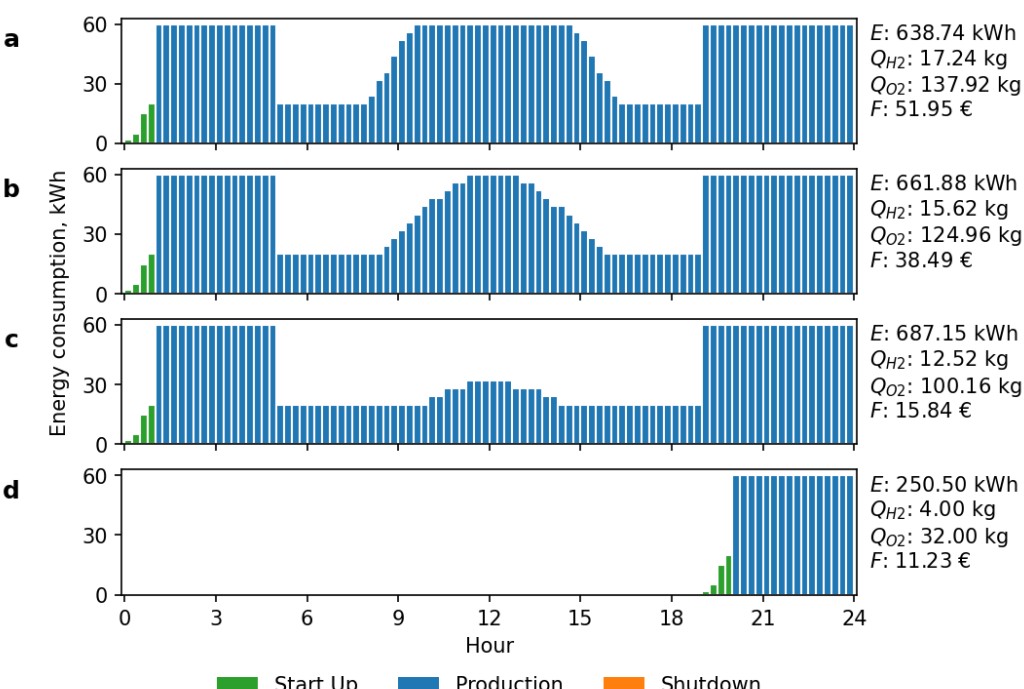

**Figure 4.** Electrolyser schedules depending on available renewable power (solar at peak): (**a**) 100 kWp (baseline), (**b**) 60 kWp, (**c**) 30 kWp and (**d**) 10 kWp.

Finally, we evaluated the total profit of the schedule depending on a range of values for various parameters (see Figure 5). We performed a number of CP-SAT model optimisations, each time changing a single input parameter. The range of the parameter values is depicted in Figure 5. We selected 10 interpolated values for each parameter range and performed total profit calculations for the optimal schedule. The results are depicted as continuous plots.

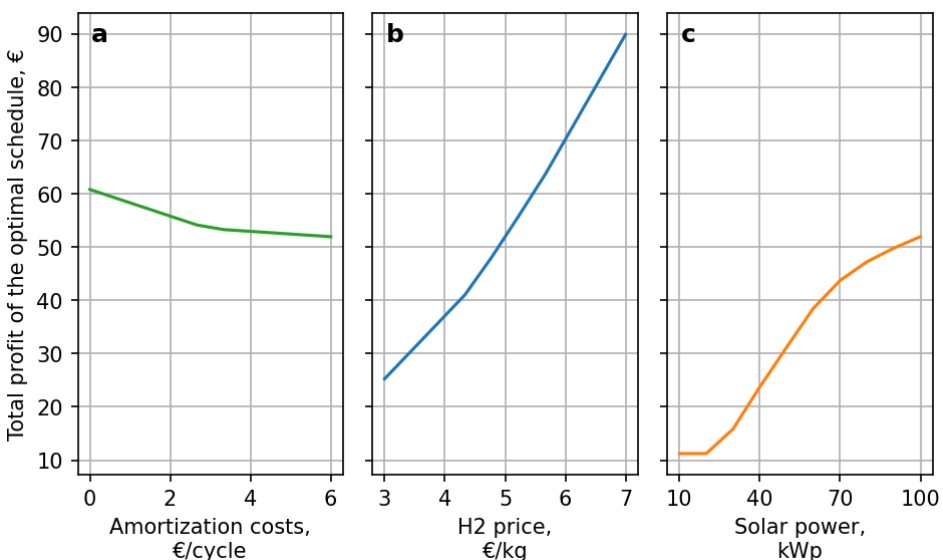

**Figure 5.** Total profit of the optimal schedule depending on input parameters of the model: (**a**) amortisation costs, (**b**) $H_2$ price and (**c**) available solar power.

Surprisingly the amortisation costs have a significant impact on the schedule itself—the schedule varies from "always on" to a highly intermittent schedule as depicted in Figure 3, but in general, it impacts only about 15% of the total enterprise. Availability of RE has a significant impact on the total profit but with a clear saturation level at electrolyser

peak power consumption. We used solar power as an RE source, which is available only during the daylight period. Adding another RE source or storing solar power for overnight use would smooth the profile of the available energy and improve the overall profit of the electrolyser.

The most significant factor is the price of the end product—hydrogen. Higher hydrogen prices make the system less susceptible to grid energy prices and their fluctuations, thus making electrolyser operation more profitable.

## 4. Conclusions

The study presents a mathematically formulated optimisation problem of the electrolyser scheduling for a 24 h time horizon. Comparison of the results between empirical (MCTS) and mathematical (CP-SAT) methods reveals several pros and cons of these two approaches.

The CP-SAT model demonstrates better performance (both in terms of time and precision) and is able to find the optimal solution, however, it has several limitations. The model uses fixed time spans across the defined time horizon. Job shop scheduling problems allow variable time spans, but they are subject to a finite number of jobs, while the electrolyser potentially has an infinite number of state transitions. Combining these two approaches for the chosen time horizon problem is a challenging task.

Another limitation of the CP-SAT model is integer variables. For example, variable production rates should be expressed as a set of values with fixed resolution rather than as a floating-point value. This peculiarity requires additional effort during the formulation of the optimisation problem and might be a decisive factor for the approach application in other domains.

The CP-SAT model by its nature is not very well scalable: adding additional parameters increases its complexity exponentially and leads to a combinatorial explosion.

The MCTS model on the other hand is more capable of handling significantly larger problems but does not guarantee optimal solutions in mathematical terms. Comparison shows that this model was only ~5–7% off from the optimal solution.

Within this study, a mathematical model for an H2-producing electrolyser has been developed and optimised using the OR-Tools suite (specifically CP-SAT solver). The suite provides remarkably effective and developer-friendly means for analysing models; the major challenge is the formulation of optimisation problems in a way suitable for the tools. The developed model can be further modified and improved by incorporating additional rules and peculiarities of the $H_2$ production process. In our case, the model is defined using Python code, which opens additional advantages of OR-Tools, namely the ability to automate repetitive parts and raise development to the metaprogramming level.

With current electrolyser technology, $H_2$ production is not profitable from grid energy (especially from fossil fuels). Common practice is to use a surplus of the RE for $H_2$ production effectively transforming it into long-term energy storage, which is later used for various applications such as mobility or small-scale electricity production. Also, this industry in its current state heavily relies on support from regulatory bodies. There are more profitable uses for the excess of renewable energy as far as $H_2$ production via methane steam reforming is available and not suppressed by regulatory bodies.

As a final thought, our intention is to further expand the electrolyser control strategy by incorporating additional parameters so that it can be implemented in a real experimental plant for validation and testing purposes.

**Author Contributions:** Conceptualisation, A.Z. and V.K.; methodology, A.Z. and A.K.; software, V.K.; validation, all authors, writing—original draft preparation, A.Z. and V.K.; writing—review and editing, V.B. and A.K.; visualisation, V.K. All authors have read and agreed to the published version of the manuscript.

**Funding:** Scientific research and publication of this article are supported by the ERA-NET Project "New technology to produce hydrogen from Renewable Energy Sources based on AI with optimised costs for environmental applications" (HydroG(re)EnergY-Env, No.112068).

**Data Availability Statement:** The original contributions presented in this study are included in the article; further inquiries can be directed to the corresponding author.

**Acknowledgments:** Acknowledgments to the collaborative ERDF project "Development of a hydrogen hydraulic compression technology for hydrogen filling stations" (H$_2$-Compression, No. 1.1.1.1/20/A/185), led by Ventspils University of Applied Sciences.

**Conflicts of Interest:** The authors declare no conflicts of interest.

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
