# Peer review of "Model for Hydrogen Production Scheduling Optimisation"

_2673-3951, doi:10.3390/modelling5010014_

Round 1

Reviewer 1 Report

Comments and Suggestions for Authors

The paper manuscript, "Model for hydrogen production scheduling optimisation", presents a simulation for optimising hydrogen production scheduling to meet the growing need for efficient and sustainable energy sources. In order to improve the overall performance of electrolysers, the research explores the integration of advanced scheduling techniques. The model proposed by the authors of the paper uses Constraint-aware Scheduling Technology (CP-SAT) to systematically analyse complex production schedules, considering factors such as production unit capacity, resource availability and energy costs. The objective of the optimization model is to improve operational efficiency and reduce production costs through the efficient tune of the electrolysis process, taking into account real constraints such as variable energy prices and the availability of renewable energy. The electrolyser schedule optimisation was performed over a 24-hour period to identify suitable schedules for water electrolysers, taking into account variable environmental factors, which can be very useful for specialists in this field.

My specific comments are:

1.    Clarity and Detailing: The manuscript would benefit from a clearer explanation of the technical terms and concepts for those readers who are not familiar with the field.

2.    Comparative Analysis: It would be useful to make a comparison with other studies or with more than one system in order to contextualise the results of the present research.

3.    Data and Methodology: To improve the reliability and reproducibility of the study, more detailed information on data sources, assumptions in the simulations and methodology is needed.

For the software version used to create the programming model, the bibliographic source must be given and it would also be advisable to present the limitations of the modules created with this software.

All the data used to calculate the figures should be summarised in tables and references to them should be given in the text.

In the economic part of the authors' study, the initial investment and maintenance costs of such a production capacity and photovoltaic park, and the period after which this investment is profitable in terms of capital expenditure (CAPEX) and operating expenditure (OPEX) should be presented for the turnover calculation to be correct.

Financial and economic indicators related to this depreciation study should have been used to present the economic results.

Mathematical relationships should be numbered and references to them should be given in the text.

4.    Environmental Impact: A broader discussion of the environmental implications of using the results of this programming method for hydrogen and oxygen production, particularly in terms of greenhouse gas emissions, would add value.

5.    Future Research Directions: For the advancement of the field, suggestions for future research based on the findings would be beneficial. Also, the article only looked at the energy provided by photovoltaic panels, but they only work when the sun is shining and at maximum parameters in sunny areas. It would be ideal to have a study in which the energy was also provided additionaly from windmills, as they can work both day and night when it is windy.

6.    Bibliographic sources: All tables and figures in the article must indicate their source (Source: author based on... or Source: [X]). 

7.    Similarity rate: The rate of similarity of the articles detected by the PlagAware software is about 17%. It would be a good idea to reduce this percentage even further.

 Comments on the Quality of English Language

The language of the manuscript is generally acceptable. However, the authors can correct minor language errors during the revision.

Author Response

Authors would like to thank the reviewer for the provided comments. Please see the attachment for our response.

Reviewer 2 Report

Comments and Suggestions for Authors This paper studies a practical scheduling model for hydrogen production process, which has practical application significance. In addition, throughout the full text, we can find that the article is very refined. But it also makes me worried. The authors proposed a CP-SAT algorithm and compared it with MCTS. I wonder if that alone is enough. Compared with the current popular meta-heuristic algorithm, does the proposed algorithm have advantages? I think this needs to be supplemented by experiments.

Comments on the Quality of English Language

Minor editing of English language required

Author Response

(The authors gave the same response as above.)

Reviewer 3 Report

Comments and Suggestions for Authors

The manuscript is interesting.The main point is to clarify the originality of this modeling approach based on two methods,namely CP-SAT and MCTS.

At the same time the advantages and drawbacks of the studied modeling have to commented.

The industrial implementation has to be discussed

.A better organization of the study is recommended. based on innovated approach considered by authors.

Comments on the Quality of English Language

Moderate  editing of English is recommended to be consideded by authors.

Author Response

(The authors gave the same response as above.)

Reviewer 4 Report

Comments and Suggestions for Authors

The manuscript presents an application of a CP-SAT (constraint programming and satisfiability) technique in a simplified model to produce hydrogen using an electrolyzer and a solar panel, in a kind of optimal on-off control problem. I consider that the content of the research is valuable and is complementary to a recent paper (2023) published by the same research group. On the other hand, some issues must be addressed before publication. The quality of the text is good, but I suggest a moderate review (in my opinion, there is a confusion in the use of commas/dots in some sentences). I have the following suggestions to the authors:

- Abstract: I consider that the abstract is somewhat generic and does not represent the content of the research. For instance, the comparisons between CP-SAT and MCTS (Monte Carlo Tree Search), discussed in the results of the manuscript, are not presented in the abstract. In the same way, the effect of the amortization cost in the schedule, also discussed in the research, is not cited in the abstract.

- Lines 59-69: The scope and motivation of the study are presented in these two paragraphs, considering the disabilities of the Monte Carlo Tree Search to solve the problem (as discussed in a publication of the same authors). I think the authors could reword the paragraphs in lines 59 to 69. The text sounds a bit personal. For instance, the statement "Authors decided to use constraint programming solving to tackle this problem" does not seem appropriate at this point of the manuscript.

- Line 108: Please define all quantities appearing in the maximization problem. For instance, Tr, G+ were not defined.

- Lines 135-152: The text (essentially, a description of some model assumptions) is the same published in the reference [11], by the same authors. I leave it up to the editor to decide whether this text should be changed.

- Lines 211-213: I consider that the sentence "The latter schedule manually entered into the MCTS model yielded exactly the same numbers, thus confirming matching calculations and better performance of the CP-SAT model" must be removed. This statement is unnecessary.

- Line 224 (Figure 2): The results presented in Figure 2 presented a "symmetrical" profile (apart from the start of the electrolyzer) for the CP-SAT technique, which is expected considering the effect of the solar radiation (maximum capacity of the solar panel at midday). On the other hand, the results for the MCTS (Figure 2(b)) do not exhibit the same type of symmetry. Is this asymmetrical profile for the MCTS an evidence that this algorithm was unable to find the global optimum?

- Lines 250-263: I am in doubt regarding the results presented in Figure 3. As far as I can see, Figure 3(c) shows a scenario with lower amortization costs (as discussed in the manuscript, representing the need to replace some parts of the equipment), in a comparison with Figure 3(a). On the other hand, Figure 3(a) presents an "always on" scenario, while in Figure 3(c) we can note that the electrolyzer is off at some moments. I consider that the authors must present a more detailed explanation regarding the effect of Cs in the obtained schedules. The authors presented only a statement in lines 103-104 ("The fixed values linked to the Start/Shutdown state cycles represent the maintenance costs attributed to internal component degradation"), but, in my opinion, this discussion should be taken up in the results section. As I understand, the effect of the Cs occurs only in the start/shutdown, which explain the "always on" policy for high values of Cs, but this information can be more clearly explained in the results. Is my understanding correct?

- Line 262: I suggest that the sentence "While cheaper..." should not start with a period.

- Line 270 (Figure 4): I cannot understand the result presented in Figure 4(d). Even considering a reduced capacity of the solar panel to 10%, why did the algorithm propose the start of the equipment without solar light? Considering the quantity -max(Es−EtPV, 0)*PtE in the objective function, sounds reasonable to me to start the equipment when EtPV is higher than zero. Is there an explanation for this situation?

- Lines 310-312: The authors concluded that "Also the CP-SAT model is limited to integer variables, which complicates optimization problem formulation. For example, variable production rate should be expressed as a set of values with fixed resolution rather than as a floating point value". On the other hand, I can't find a discussion regarding the differences between the implementation of the algorithms in the results section that supports this statement. In my opinion, this is not a conclusion, but a characteristic of the algorithm.

- Lines 313-314: The authors stated that "The MCTS model on the other hand is capable of handling significantly larger problems, but does not guarantee optimal solutions in mathematical terms". Again, I consider that this is not a direct conclusion of the research. I suggest that the authors must improve this statement.

- Lines 322-323: Once again, I consider that the statement "...and the optimization rules can be easily implemented in the code" cannot be supported by the results presented in the manuscript.

- Line 331: The authors cited H2 production by steam reforming in the last line of the manuscript, but without any previous information regarding this process (for instance, in the introduction of the manuscript). Furthermore, the statement "Also there are more profitable uses for the excess of renewable energy as far as H2 production via methane steam reforming is available and not suppressed by regulatory bodies" is not supported by the results obtained or by references presented in the research. In my opinion, this statement even detracts from the interesting results obtained by the authors, since - as pointed by the authors - the profitability of the process is essentially depending on the technology and tax incentives/environmental regulations.

- Finally, I consider, generally speaking, that the model must be more clearly explained (even considering that a detailed description is presented in a previous work of the same group), as well as the results (some sentences are too concise).

Comments on the Quality of English Language

The quality of the text is good, but I suggest a moderate review (in my opinion, there is a confusion in the use of commas/dots in some sentences).

Author Response

(The authors gave the same response as above.)

Round 2

Reviewer 1 Report

Comments and Suggestions for Authors

My specific comments are:

1.  Clarity and Detailing: The modifications are fine.

2. Comparative Analysis: Comparison with other studies by other authors or across different systems would be useful for contextualisation of the results of the present research.

3. Data and Methodology: The manuscript has been the subject of improvement sufficiently..

4. Environmental Impact: I hope that the authors will succeed in writing an article that highlights the environmental implications of using the results of this programming method for hydrogen and oxygen production, particularly in terms of greenhouse gas emissions.

5. Future Research Directions: The addition of explanations in the article is fine, but the values for the other renewable energy sources could have been included in the calculations for comparison of value creation.

6. Bibliographic sources: All tables and figures in the article must indicate their source (Source: author based on... or Source: [X]).

I understand that all figures and tables are the result of the authors' work on the basis of the results of the study, but in the binding they are the result of the use of a software product. The indication of the sources is not redundant, as it is a reference to the non-originality of the figures; they are the result of the presentation through the use of interfaces.

7. Similarity rate: The rate of similarity of the articles detected by the PlagAware software is about 17%. It would be a good idea to reduce this percentage even further.

Author Response

Thanks once again for your comments. Please find our replies in the attachment.

Reviewer 3 Report

Comments and Suggestions for Authors

The article is well written.My recommendations are as follows:

1Industrial implementation of the approach presented in this article

2 list of references to be completed

Comments on the Quality of English Language

Minor editing of English is required.

Author Response

Authors would like to thank the reviewer for the time and comments.

1 - Peculiarities of the industrial implementation are out of scope of current research, but can be considered for the future research studies.

2 - References style is changed from Surname, Year to [#]